# Encapsulation of Active Substances in Natural Polymer Coatings

**DOI:** 10.3390/ma17112774

**Published:** 2024-06-06

**Authors:** Emma Akpo, Camille Colin, Aurélie Perrin, Julien Cambedouzou, David Cornu

**Affiliations:** IEM, Université de Montpellier, CNRS, ENSCM, F-34095 Montpellier, France

**Keywords:** biopolymers, protection, ionic gelation, hydrogels, formulation

## Abstract

Already used in the food, pharmaceutical, cosmetic, and agrochemical industries, encapsulation is a strategy used to protect active ingredients from external degradation factors and to control their release kinetics. Various encapsulation techniques have been studied, both to optimise the level of protection with respect to the nature of the aggressor and to favour a release mechanism between diffusion of the active compounds and degradation of the barrier material. Biopolymers are of particular interest as wall materials because of their biocompatibility, biodegradability, and non-toxicity. By forming a stable hydrogel around the drug, they provide a ‘smart’ barrier whose behaviour can change in response to environmental conditions. After a comprehensive description of the concept of encapsulation and the main technologies used to achieve encapsulation, including micro- and nano-gels, the mechanisms of controlled release of active compounds are presented. A panorama of natural polymers as wall materials is then presented, highlighting the main results associated with each polymer and attempting to identify the most cost-effective and suitable methods in terms of the encapsulated drug.

## 1. Introduction

Encapsulation has been a very active area of research in recent years. Encapsulation technologies and formulations have been investigated by the scientific community for various applications. It is an effective way of protecting and preserving active substances from the conditions of the surrounding medium conditions such as moisture, reactive chemicals of the medium, and UV light, among others. It can also be used to deliver drugs to a target site. A number of parameters are important in the design of materials for a specific encapsulation process, including pH, salt used, temperature, and pressure. The quality of the encapsulation is also influenced by the nature of the active ingredient, the nature of the wall material, and their interactions. A material used as a coating should have high functional rheological properties, and for most applications, should be edible, biodegradable, compatible with the active ingredient, cost-effective, and provide optimal stability during and after the manufacture of the encapsulated products [1,2]. Several molecular compounds have been investigated and used as encapsulated substances. Lipids have been used to entrap organic compounds in a self-assembling liposome [3,4,5]. Polymers are also relevant candidates as drug matrices. They can be of synthetic or natural origin. Even if some synthetic polymers show good biocompatibility and biodegradability properties, they might present a restrained interest considering the petrochemical origin of most of them. Biopolymers have been studied for the design of hydrogels capable of encapsulating the core compounds, giving polymeric networks with a versatile behaviour. Encapsulation has been used in various industries, such as the flavour industry to preserve the organoleptic properties of molecular compounds [6] and in the pharmaceutical industry for the encapsulation of anticancer drugs [7,8,9].

In the context of environmental conservation, bioprotection solutions have recently been developed to design greener tools capable of controlling threatening or pestilential organisms [10,11,12], pathogens, weeds, or pests to provide human [13] and environmental benefits [10,14,15]. Biopesticide alternatives are natural approaches to combat a target by using natural phenomena already observed in nature. Natural molecular substances and biocontrol agents, which require the use of living organisms, are among the biopesticide alternatives. These new approaches involve sensitive components such as natural molecular substances, e.g., plant-derived substances [16] and proteins [17], and biocontrol agents, e.g., microbial biocontrol agents vectored by bumble bees [18]. Encapsulation for such applications in the agrochemical industry requires low-cost coating materials, an environmentally friendly composition, a cost-effective encapsulation technique, and an easy scale-up of the manufacturing process to fabricate large quantities of products. The current challenge is to develop new agrochemical treatments that are environmentally friendly, cost-effective to maintain competitiveness in a highly competitive industry, user-friendly, and capable of widespread application for rapid treatment of large areas [10,11]. Biopolymers have already been investigated for the microencapsulation of beneficial microorganisms capable of combating plant pathogens and developing new agrochemicals [19].

This review provides an overview of the knowledge already gained on encapsulation in biopolymers of different active substances for various types of applications. It is aimed at identifying strategies for the delivery of active substances based on their nature, physico-chemical properties, and characteristics of both the storage and the delivery sites.

## 2. The Concept of Encapsulation

Encapsulation is the process of creating a functional barrier between a core material and its surrounding environment to prevent chemical reactions and physical interactions, and to maintain the biological, functional, and physicochemical properties of the core material [20,21,22]. For application in the food industry, encapsulation provides a protective shell for the food colourant, preventing it from easily migrating through the product, as is the case with conventional colourants [23]. Encapsulation is also a strategy that can be used to protect highly sensitive components such as essential oils [3], e.g., Pimento essential oil, which exhibits antifungal activity, especially against dangerous pathogenic and toxigenic fungi, was microencapsulated using chitosan and κ-carrageenan [4,24]. Encapsulation of oils has been reported to improve the oxidative, thermal, moisture, and light stability, shelf life, and biological activity of oils [20,25]. Physical or chemical encapsulation processes produce beads with sizes ranging from a few nanometres for nanoencapsulation, to a few millimetres for microencapsulation [11]. Encapsulation can be achieved in particular by using polymers as wall materials. Different forms of capsules can be designed with more or less complex architectures (Figure 1). The common architectures are the spherical simple microcapsules, for which the core material is surrounded by a layer of material, and the microsphere, for which the core material is trapped and dispersed in a polymer matrix. However, the structure of the microsystem can also be irregular depending on the encapsulation process used [20,26].

## 3. Technologies of Encapsulation

Several technologies have been developed for achieving encapsulation. These encapsulation methods can be classified into physical methods, chemical methods, and physico-chemical methods. Technology screening depends on the physico-chemical properties of the carrier and of the core ingredient, the particle size target, and the encapsulation efficiency. The method of encapsulation also influences the sensitivity of the capsules to the surrounding medium and the kinetic release of the active ingredient.

A panorama of the encapsulation methods is presented in Table 1. In this review, we focus on three conventional methods—spray-drying, extrusion, and coacervation—, which are commonly used for microencapsulation. These methods have been used in particular for the encapsulation of active ingredients such as plant growth-promoting bacteria [11].

### 3.1. Spray Drying

Spray-drying is widely used to microencapsulate molecular and enzymatic actives, such as carotenoids, lipids, or enzymes [40,41,42]. Easily implemented on a large scale [20], spray drying is a continuous process that converts various liquids, solutions, slurries, dispersions, pastes, or even melts into solid particles with adjustable sizes, shape distributions, porosities, densities, and chemical compositions (Figure 2) [43]. The feed, containing the active ingredient and the wall materials, is atomised in the drying chamber, where the water in the droplets formed is immediately evaporated by contact with the hot air inside the chamber. The formed particles are then separated from the drying air by a recovery cyclone [44]. This fast and inexpensive method is particularly suitable for capturing enzymes. The short contact time of the heat with the formed particles makes spray drying very attractive for enzyme encapsulation [45]. However, the encapsulation of active ingredients such as bacteria is potentially altered by the drying step [4].

### 3.2. Extrusion

Manually or automatically, the feed material is extruded through a syringe into a cross-linking bath to obtain and stabilise the microbeads. The extrusion technique has been used to develop mixed locust bean gum and alginate microbeads through the ionotropic gelation method. A specific amount of the locust bean gum/alginate polymer blend and the drug were first dissolved in distilled water. The feed material was then dropped through a needle into a beaker containing an aqueous calcium chloride solution (Figure 3) [7]. A high-voltage generator can be used to oppositely polarise the syringe and the calcium bath, thereby orienting the droplet flow and regulating the droplet size, thereby influencing the size of the beads. Extrusion is an economical and straightforward encapsulation process that does not require the use of harmful solvents [46]. In addition, encapsulation can be used under anaerobic and aerobic conditions. The size of the beads depends on the distance between the cross-linking bath and the syringe, the viscosity of the mixture, the polymer concentration, and the diameter of the syringe needle [47]. The two main problems with extrusion methods are the difficulty of large-scale application and the slow bead formation [11]. This technique is particularly suitable for the encapsulation of living cells. In cell-based therapy, alginate (1.5% *w*/*v*)/agarose (0.1% *w*/*v*) beads were chosen to encapsulate, protect and act as a vehicle to transport the stem cells. Delivery was by bursting release of the stem cells from the beads upon reaching the active site [48].

### 3.3. Coacervation

In recent years, the coacervation method has been investigated for the microencapsulation of oils. Coacervation is the oldest and most widely used encapsulation technique [20]. This relatively simple method can be compared to a modified emulsification technique. The process involves the separation of the hydrocolloid from the primary solution and subsequent aggregation into a distinct liquid phase known as “coacervation” [49]. In the first step of the coacervation process, the core material is suspended in a continuous liquid phase. In the second step, the coating material is added to the two-phase system. The third and fourth steps involve gelation and solidification of the microcapsule wall around the core (Figure 4) [50].

The wall formed by the polymer plays an interesting role in protecting the microencapsulated oil and exhibits excellent controlled release. In addition, this low-cost method can be used to produce microcapsules on an industrial scale. The encapsulation of protein through polypeptide complex coacervation offers an efficient method for controlling extended release in response to pH changes [51]. Coacervation processes have been classified into simple and complex coacervation [20]. In simple coacervation, the polymer is deposited around the core material by salting the polymer with the addition of electrolytes, or by desolvation of the polymer with the addition of a water-miscible non-solvent, or by playing with the temperature of the liquid environment [50]. In complex coacervation [52], microcapsules are formed by the interaction of two or more oppositely charged colloids or polyelectrolytes, usually proteins and polysaccharides [40]. This process is influenced by parameters such as pH, the isoelectric point of proteins, the ionic strength:polysaccharide ratio, the total concentration of biopolymers, the type of core material, and the core:wall ratio. Stirring speed also plays a key role in controlling the size of the coacervates formed. One of the advantages of this approach is the overall higher encapsulation efficiency. The combination of gelatin and arabic gum is a standard for complex coacervation. The attraction between type A gelatin and gum arabic is observed at a pH below 9. The coacervation mechanism between gelatin and gum arabic is explained by the electrostatic interaction between the positively charged ammonium moieties of gelatin and the negative charges of gum arabic chains. Gelatin and gum arabic, when exposed to electrostatic interactions, form a coacervate layer, which hardens during the gelatin cross-linking process [50,53,54].

## 4. Encapsulation in Micro- and Nanogels

In order to form micro- and nanoparticles using the described biopolymers, different manufacturing strategies are used, depending on the chemical structure and physicochemical properties of the gums. The formation of a gel is of interest in order to preserve the integrity of the active substance in the capsules, such as enzymes or probiotics, among others. The preparation method can have a significant impact on the final properties of the hydrogels, a critical factor to be considered, especially when evaluating drug release kinetics.

### 4.1. Mechanisms of Formation of Hydrogels

#### 4.1.1. Ionic Cross-Linking

Ionic cross-linking is a method of gel formation for some gums that can interact with di- or trivalent cations, such as gellan gum or alginate, or an artificial polymer such as acrylic acid (Figure 5). The cations interact with the negatively charged parts of the polysaccharide, forming bridges and initiating the formation of a gel structure. The active ingredients are trapped between the cross-linked polysaccharide chains. Reversible and mild, ionic cross-linking potentially makes the obtained products fully biocompatible according to the screening of polymers and ions. It is a complex phenomenon, difficult to describe, and highly dependent on the temperature, which affects the behaviour and the structural organisation of the biopolymer chains in solution. The study of the effect of various divalent metal cations has highlighted the significant impact of ionic strength and hydrogen bond interactions on the mechanical properties of bioinspired Montmorillonite-alginate hybrid films. Alginates have shown varying degrees of affinity depending on the binding divalent cations: Mn^2+^, Co^2+^, Ni^2+^, and Ca^2+^ and exhibited strong ionic cross-linking with Ba^2+^, Cd^2+^, and Cu^2+^ [55].

A classical realisation of ionic crosslinking is found in the case of alginates, which will be described in more detail in Section 6.1.1. In sodium alginates, the sodium ions are usually replaced by calcium cations and the gel formation process consists of a few steps. In the initial stage, the monocomplexes are formed, and in further steps, they transform into dimers and multimers, which is commonly known as the “egg box” structure [57]. According to another theory, the crosslinked polymer chains that form the dimer are not parallel but approximately perpendicular to each other. This results in a tilted “egg box” structure [58]. Similar properties have been described for gellan gum [59].

#### 4.1.2. Covalent Cross-Linking

Covalent cross-linking is an irreversible way of bridging the adjacent polymer chains. The process is irreversible and the bonds obtained are generally insensitive to pH changes (Figure 6) [60]. The cross-linker used must contain at least two functional groups capable of reacting with the hydroxyl groups in gum molecules or other functional groups in the polymer chains to link the two chains [61]. Glutaraldehyde and sodium trimetaphosphate are known to be the most commonly used cross-linkers, but dialdehyde is said to be toxic, limiting its use in pharmaceutical technology [56,61]. Less harmful cross-linking agents have been proposed, such as genipin [62] or citric acid [63].

#### 4.1.3. Polyelectrolyte Cross-Linking

This process is commonly used to produce microspheres and microcapsules. For example, the polyelectrolyte complex can be deposited at the interface between the oil droplets and the aqueous medium, forming microcapsules [51]. Gelling occurs through the interaction of two oppositely charged polymers (Figure 7) [64]. The negatively charged residues of one polymer interact with the positively charged residues of the second polymer [56]. Inter-complex aggregation is strongly dependent on the ratio of cationic to anionic groups. In the case of a non-stoichiometric ratio, the product is water soluble. The formation and subsequent behaviour of the polyelectrolyte complex are influenced by pH, electrolyte concentration, and the mixing order of the specific compounds [56].

#### 4.1.4. Polysaccharide and Drug Conjugation

In this technique, active molecules are covalently attached to the polysaccharide backbone. The spontaneous process of self-assembly is facilitated by the attachment of hydrophobic drugs to the hydrophilic structure of the gum chains (Figure 8). The polymer backbone is used as a shuttle to transport the drugs to the active site [65]. The bonds between the active molecules and the carrier gum must be hydrolysed or cleaved under specific conditions and the drug must be released in its original form [66]. A biodegradable linker can also be used to connect the two components [67,68].

#### 4.1.5. Self-Assembly

The self-assembly mechanism is observed in the case of amphiphilic molecules that adopt a spatial arrangement to minimise the contact between their hydrophobic parts and the surrounding hydrophilic environment. The hydrophilic part is located at the interface between the hydrophobic and polar phases (Figure 9). The driving force for aggregation is the hydrophobic effect, which minimises the interfacial energy between water molecules and hydrophobic domains [60]. The polymeric gums can be modified with hydrophobic moieties introduced into the structure to initiate molecular self-assembly [69]. Parameters such as particle diameter, zeta potential, or loading efficiency can be modulated by adjusting the polymer chain length and the size of the hydrophobic residues [56,60].

### 4.2. Hydrogel Additives

#### 4.2.1. Chelating Agents

Chelating agents are incorporated into the microgels to improve their functional performance. The chelating agent, ethylenediaminetetraacetic (EDTA), was used as a control agent for the formation of alginate beads under mild conditions [70]. The probiotic, Lactobacillus rhamnosus ATCC 53103, was encapsulated in a pH-responsive hydrogel based on an EDTA calcium alginate (EDTA-Ca-Alg) system. The pH-responsive system exploited the reversibility of the EDTA-calcium complexation reaction. The probiotics were protected in the acidic environment of the stomach but released in a neutral environment such as the small intestine due to the release of calcium ions by altering the pH [71,72].

#### 4.2.2. Buffers

Small ions, such as protons and hydroxyl ions can easily penetrate biopolymer microgels and alter their internal pH. Buffers or antiacids with low solubility in water can be incorporated into the microgels to control the internal pH, such as CaCO_3_, Ca(OH)_2_, Ca_3_(PO_4_)_2_, MgCO_3_, Mg(OH)_2_, ZnCO_3_, Zn(OH)_2_, and Zn_3_(PO_4_)_2_ [73]. Mg(OH)_2_ particles have been incorporated into the formulation of alginate beads to improve viability under gastric conditions [74]. Dissolution of the antiacid under acidic conditions released hydroxyl ions. As a result, the pH inside the hydrogel remained neutral, increasing the resistance of the probiotic cells under unfavourable acidic conditions [73,74].

#### 4.2.3. Hydrophobic Substances

The functional properties of biopolymer microgels can be modified by incorporating hydrophobic substances such as lipid droplets, vesicles, or non-polar protein particles. These substances can act as a physical barrier, preventing the diffusion of water-soluble substances into and out of the microgels. In addition, they can be used to solubilise hydrophobic substances or act as a source of probiotic nutrients. Probiotics have been encapsulated into hydrophobic/alginate beads. The results of the study proved that cocoa butter as an additive for alginate beads acts as a protective agent on *L. rhamnosus* in simulated gastrointestinal tract acidic conditions [71,75].

## 5. Mechanisms of Active Compound Release

Microcapsules are often semipermeable and approximately round systems that provide controlled release properties under specific physical, chemical, or mechanical conditions [76]. Smart polymer systems have been developed as solid packaging to encapsulate anti-inflammatory and antibiotics drugs, such as Diclophenac, Metronidazole, and Indomethacin, as a promising route for controlled release into the human body [77]. Swelling, diffusion, and a combination of both erosion and degradation have been identified as the main mechanisms governing drug release from particles such as chitosan nanoparticles for the controlled release of enzymes [4]. Figure 10 summarises the various phenomena that eventually lead to the release of active compounds.

### 5.1. Fragmentation

The fragmentation release occurs when external conditions, such as mechanical pressure, shear, pH changes, or others, weaken the polymeric structure and separate it into several parts containing a ratio of active substances (Figure 10B) [77].

### 5.2. Diffusion and Swelling

In diffusion-controlled release, the active compounds permeate through the porous structure of entangled polymer chains forming a barrier that slows down the release rate [78]. The diffusion is characterised by a typical drug release profile, featuring an initial fast release, also called the burst effect [79]. Diffusion release is often induced by the swelling mechanism. In the swelling mechanism, the polymer network is progressively detangled by the interaction with solvent molecules such as water or other molecules from the surrounding aqueous medium [80]. The swelling rate depends on the physico-chemical properties of the wall material and its environment, such as temperature, ionic strength, and pH (Figure 10C,D).

### 5.3. Dissolution and Erosion (Degradation)

Dissolution and erosion are interrelated phenomena impacting the integrity of the polymer matrices (Figure 10E,F). Erosion is a multicausal phenomenon involving swelling, diffusion, and dissolution of the polymer chains. It can occur by two different mechanisms: heterogeneous and homogeneous erosion. Homogeneous erosion occurs at the same rate throughout the wall material, whereas heterogeneous erosion occurs from the surface to the inner core of the capsule [77]. Dissolution and erosion follow a zero-order mechanism, meaning that the drug release process is independent of the drug concentration and rather depends on the dissolution or degradation of the polymer chains over time [81,82]. A controlled drug delivery system for oral administration has been developed, designed to exhibit a delayed burst release profile. This system aims to release pentoxifylline, theophylline, and theobromine at a specific time through rapid erosion, which is finely adjusted by blending various carboxylic acids and polyvinyl alcohol [81].

### 5.4. Release under External Conditions Control

Based on the chemical structure of the polymer and the surrounding environmental conditions, particles can exhibit swelling and dissolution (Figure 10F). The smart behaviour of multi-responsive microgels mainly depends on pH and temperature (Figure 11).

Poly-N,n-isopropylacrylamide (PNIPAM) is one of the most prominent examples of stimuli-responsive microgels, which are thermosensitive particles. The copolymerisation of PNIPAM with acrylic acid was achieved to tailor the stimuli-responsive behaviour by adding a pH response. The radius of the spherical nanostructure changes due to deprotonation or protonation of the chemical functions of the repeating units (Figure 6) [83].

The release of active compounds is a complex process combining all of the above mechanisms. The stability of the capsule and the release of the drug are highly dependent on the surrounding parameters. In vitro tests can be performed to evaluate the stability of the system and the protective effect on the drug. Experimental tests were carried out to evaluate the stability of the microcapsules made from whey proteins and gellan gum encapsulating flaxseed oils and flaxseed hydrolysate proteins. In the absence of enzymes, the low pH did not alter the morphology and the monomodal size distribution of the microspheres. On the contrary, in the in vitro test, which simulates the conditions of intestinal digestion, a large number of uncoated oil-in-water emulsion droplets were observed in the system after only 15 min in the presence of the digestive enzymes. The addition of digestive enzymes destabilised the gelled layer and triggered the release of the emulsions containing flaxseed oil and flaxseed proteins [84].

## 6. Wall Materials: Natural Polymer Coatings

Polymers are macromolecules made up of multiple repeats of patterns containing one or more units called monomers. These large molecules can be divided into three categories according to their source: natural, synthetic, and semi-synthetic, and all of them can be used as wall material for a variety of active compounds [4]. Derived from plants, animal waste, or bacteria, biopolymers are popular as core materials because of their biocompatibility, biodegradability, natural origin, and non-toxicity. In particular, biopolymers are used as drug carriers or scaffolds for cell implantation in regenerative medicine. One of the drawbacks of the use of biopolymers is the variability in the chemical structure, such as the sequence and ratio of repeating units, according to the origin of extraction, so that the encapsulation performance can be affected by this variable character of the molecular structure. Biopolymers can be divided into two main categories: proteins and polysaccharides. Biopolymer blends are an effective formulation strategy to control and improve the stability of capsules, increase the amount of active substance loading in the polymer matrix, and accurately target its delivery destination.

The choice of wall material and encapsulation technique for microcapsules depends on the nature of the core materials, the use of the beads, and the processing conditions involved during the delivery process. In the food, pharmaceutical, cosmetic, biomedical, and agrochemical industries, microencapsulation with biopolymers is used to encapsulate different core materials such as enzymes, probiotic bacteria, oils, flavourings, and bioactive molecules. Beneficial microorganisms have also been encapsulated in natural coatings for the biological control of plant pathogens [4,11].

A combination of protein and polysaccharide was investigated as a wall material to improve the bioavailability and preserve the anti-inflammatory, antioxidant, and hypertensive properties of flaxseed oil and flaxseed proteins hydrolysate. The encapsulation of flaxseed oil and flaxseed protein hydrolysate, in a formulation of gellan gum and of whey protein isolate has been designed to pass through the small intestine and resist simulated gastric conditions. Beads between 50 µm to 55 µm were obtained by formulating of dense biopolymer network from the extrusion of the oil-in-water emulsions into a 0.56% calcium chloride bath [85].

### 6.1. Polysaccharides

Polysaccharides are a class of biopolymers. Their structure consists of a repetition of sugar units. Polysaccharide coatings form a barrier that limits the exchange between the core capsules and the environment [86]. Encapsulation with this type of biopolymers prevents the effect of oxygen on the active ingredients. In the case of essential oil encapsulation, the polysaccharides contain the oils, preventing the spread and the loss of the oily core material [50]. Due to their hydrophilicity, polysaccharides have the disadvantage of being permeable to moisture. The wide range of possible polysaccharides results in a wide range of encapsulation systems. Polysaccharides such as alginate or κ-carrageenan are polyelectrolytes; depending on the pH of the medium macromolecular compounds can globally adopt a negative charge due to carboxylate and sulphate groups attached to the macromolecular structure. In addition, the chemical structure of polysaccharides can be chemically or physically modified to offer new possibilities and optimise the encapsulation process.

#### 6.1.1. Alginates

Considered non-toxic and inexpensive, alginates are one of the most important biopolymers. They are obtained from three species of brown algae: Macrocystis pyrifera, Laminaria digitata, and Laminaria saccharina. This heteropolysaccharide consists of β-D-mannuronic acid and α-L-guluronic acid subunits (Figure 12). Its chemical and physical properties depend on the arrangement of these monomers, the ratio of each monomer, and the molecular weight [87,88].

Unlike sodium alginate, which is water soluble, alginate chains are substituted with divalent or multivalent counterions precipitate. The binding between the subunits of saccharide and the calcium ions results in the formation of the strong and stable three-dimensional polymer matrix, also known as a gel (Figure 13) [89].

Due to the carboxylate functions, the alginate polymer matrix undergoes morphological and chemical changes at different pH values. Above pKa 4.4, electrostatic repulsion leads to expansion of the matrix, in contrast to pHs below pKa 3.4, which promote shrinkage of the cross-linked network [91,92]. Pure alginate capsules are characterised by low mechanical properties, which can be improved by blending alginate with other biopolymers [87]. In the food industry, alginate microparticles have been used as carriers for probiotics of the genus Lactobacillus, including L. plantarum, L. acidophilus, and L. reuteri. The encapsulation process by extrusion not only protected the probiotics against unfavourable conditions in the digestive tract but also improved their viability and increased their survival rate [93].

#### 6.1.2. Pectins

Pectins (Figure 14) are characterised by a complex mixture of anionic water-soluble polysaccharides [94]. Pectins are extracted from the cell walls of fruits such as lemon peel and apple pomace. The main component of pectins is homogalacturonan; the ratio can reach 65% depending on the type of extraction. This linear polysaccharide is composed of d-galacturonic acid units [4].

The carboxyl groups present in the sugar units can be partially methylesterified and partially or completely neutralised with a counter-positive ion [25]. Depending on the number of carboxyl groups esterified pectins are divided into two groups: high methoxyl pectins with a degree of esterification of more than 50%, and low methoxyl pectins with less than 50% of methylesterified carboxyl groups. The higher the degree of acetylation of pectins, the stiffer the pectin structure. The other polymer categories are rhamnogalacturonan-I and rhamnogalacturonan-II, both characterised by a much more complex chemical structure than homogalacturonan [95].

Pectin can be converted into hydrogels, forming a flexible polymer network. In an acidic environment (pH below 3) or with high concentrations of co-solutes, high methoxyl pectin gels via self-assembly due to interactions with methyl groups and hydrogen bonds. In contrast, low methoxyl pectins gel at pH 3 to 7 in the presence of positively charged species such as calcium ions, which interact with carboxylate groups. For encapsulation, pectin with a low degree of esterification is preferred due to its low molecular weight and the formation of a water-insoluble cross-linked polymer, calcium pectinate, in the presence of calcium ions [96,97].

Nevertheless, the swelling ability of pectin hydrogels and the associated larger pore size under physiological conditions could be a limitation for a certain number of lower molecular weight bioactive compounds. In addition, pectin capsules were ineffective in efficiently entrapping hydrophobic compounds due to their hydrophilic structures. However, pectin showed an ability to protect the active compounds when the microcapsules were exposed to the harsh in vivo conditions of the stomach and upper gastrointestinal tract. In addition, low methoxyl pectin exhibited high adherence to mucosal surfaces, due to the interaction of the carboxyl groups of the pectin with functional groups on the mucosal wall such as hydroxyl, amide, carboxyl, or sulphate groups. The mucoadhesiveness of pectin makes it interesting for the nasal treatment of localised diseases and for the specific delivery of drugs [98].

#### 6.1.3. Starch: Amylose and Amylopectin

Mainly extracted from potatoes, maize, and wheat, starch is a biopolymer composed of two non-ionic polysaccharides: amylose and amylopectin [99]. It is an inexpensive and non-toxic alternative for encapsulating molecules or microorganisms such as bacteria, fatty acids, and active ingredients for pesticide, herbicide, and fungicide applications [11]. Amylose (Figure 15) is a natural polysaccharide composed of 20 to 20,000 α-D-glucose units [100,101]. Amylopectin is a branched polysaccharide composed of α-D-glucose units, consisting of approximately 2 million units with a backbone of branched short chains between 20 and 30 α-D-glucose units. The ratio is between 20 and 30% for amylose and 70–80% for amylopectin, the two ratios depend on the origin of the starch [100].

Amylopectin (Figure 16) and amylose are able to withstand acid pH. Amylose is correlated with pasting and gel properties, whereas amylopectin is correlated with firmness. Insoluble in cold water, starch and amylopectin can be dispersed in water between 80 °C and 100 °C to form a gel or a viscous liquid by gelatinisation. At around 100 °C, the starch granules absorb water and swell, the polymer chains are mobile and dispersed in the water. Below 5%, a non-gelling starch mixture is obtained. Above 5%, starch, or amylopectin forms a hydrogel in water due to the proximity and interaction between the polymer chains [102].

Starch can be modified to produce starch-based coatings because of the unfavourable chemical and physical properties of native starch, such as insolubility in cold water, and stability under different pH and temperature conditions [100,101]. Native starch granules are modified by various modification methods: chemical derivatisation (etherification, esterification, acetylation, oxidation …); an enzymatic method consisting of hydrolysis with amylase; or physical treatments such as superheating or dry heating, which shortens the chain length or modify the chemical properties of the polysaccharide chains [11,99,101]. Non-native starch macromolecules are used to improve properties and functionality such as solubility, texture, viscosity, thermal stability, water solubility, hydrophobicity, amphiphilicity, emulsifiability, digestive resistance, film-forming ability, thermal stability, and adsorption capacity [11].

A starch mixture was used for bead formation. The encapsulation of probiotics was achieved by combining starch with alginate, with the latter being sensitive to calcium ions. The beads containing alginate and starch showed a better encapsulation efficiency (77%) than pure alginate beads (64.4%) [103]. Encapsulation with pure starch is generally performed by spray drying due to the non-ionic nature of starch. During the process, the rapid drying step provides a solid shell around the core material to protect it. Starch should be functionalised or combined with another biopolymer that can more easily form a hydrogel [102].

#### 6.1.4. κ-Carrageenan

Carrageenans are classified into six types, κ-, ι-, λ-, μ-, ν-, and θ-carrageenans based on their structure. They are sulphated polygalactans with a proportion of ester sulphate groups between 15 and 40%. Carrageenans are extracted from red seaweed and have various beneficial effects due to the variability of their structure and their physico-chemical properties [104]. Κ-carrageenan is one of the most popular carrageenans. It is used as a gelling, stabilising, and thickening agent in the food and cosmetic industries, but also in the medical and pharmaceutical fields for drug delivery applications due to its high gelling ability [105].

Κ-carrageenan has a linear anionic structure formed by an alternation of 3,6-anhydro-galactose and d-galactose units (Figure 17) The physico-chemical properties are influenced by the number and the position of the ester sulphate groups on the polysaccharide chain. The content of 3,6-anhydro-galactose also influences the properties of the polymer [105]. Moderately soluble in water at room temperature, κ-carrageenan can be dissolved between 40 and 45 °C [106], up to 60 to 90 °C for high molecular chains [107]. The behaviour of κ-carrageenan is largely influenced by the nature of the surrounding ions in aqueous solution. Gelation can be induced by temperature change, from 40 °C to room temperature, in the presence of monovalent ions such as potassium to stabilise the network of κ-carrageenan chains and prevent swelling damage to the structure [106]. The temperature (between 50 and 100 °C) required to properly solubilise κ-carrageenan can lead to the death of certain bacteria during the preparation steps of the hydrogel. In addition, κ-carrageenan hydrogels have low structural stability under physiological conditions. Κ-carrageenan has been used for the encapsulation of enzymes [4], microorganisms such as bacteria, and active compounds such as tea polyphenol compounds via spray drying, emulsion, ionic gelation, and other methods. Κ-carrageenan encapsulation provided an efficient protective barrier for tea polyphenol compounds, improving bioavailability due to better compatibility with acidic environments than other biopolymers, such as alginate [108].

Encapsulation of *Salmonella phage SL101* in alginate and κ-carrageenan microbeads by the extrusion method and ionic gelation process showed a protective effect, safe-guarding the phage from inactivation under low pH conditions while maintaining its release and lytic activity over time. Alginate microbeads provided limited phage protection in the gastric environment. The combination of alginate and carrageenan improved phage protection against acidic conditions. In general, the composite microcapsules significantly enhanced phage viability in the gastric environment, although different ratios of alginate and κ-carrageenan showed inconsistent protective efficacy (Figure 18) [109].

#### 6.1.5. Gellan Gum

Gellan gum is a linear heteropolysaccharide characterised by a pattern constituted of four sugar units: glucoronic acid, rhamnose, and two glucose units [110]. It is secreted by the bacterium Pseudomonas elodea during aerobic fermentation. Gellan gum is divided into two categories: high (Figure 19) and low (Figure 20) acyl gellan gum, the latter being also known as deacylated gellan gum and obtained by the alkaline or acid hydrolysis of gellan gum chains.

This biopolymer is a low-cost raw material, commonly used as a thickener, stabiliser, binder, or gelling agent in the food and cosmetic industries, for tissue engineering applications in the medicine field, and also in the pharmaceutical field for drug delivery applications [111]. The dissolution of gellan gum requires a temperature above 40 °C. The hydrogel formation of gellan gum can be induced by temperature variation. The gel setting temperature is between 30 °C and 40 °C. Multivalent ions promote gel formation and induce effective ionic cross-linking by shielding the electrostatic repulsion between the carboxylate moieties of the polymer chains. The process induces an intimate aggregation of the ordered anionic double helices, which transform into a three-dimensional hydrated network [112]. While native gellan gum can be used as a raw material for encapsulation applications, modifying the structure of gellan gum offers improved properties for encapsulation. The structure of the polysaccharide can be chemically or structurally modified. These changes to the molecular weight of the biopolymer can be achieved by alkaline or acid treatment of the polymer chains [111]. Gellan gum has been used as a polymer matrix for the micro- and nanoencapsulation of enzymes, bacteria, and among other bioactive compounds, thanks to the extrusion method combined with ionotropic gelation [110].

The biopolymer beads made from sodium alginate and gellan gum with the addition of the surfactant decyl glucoside were obtained using the ionotropic gelation process. The beads prepared from a mixture of gellan gum and sodium alginate exhibit better stability in solutions with pH values ranging from acidic to alkaline than the alginate beads (Figure 21). The beads retained their shape, integrity, and functionality even after 24 h of incubation [113].

#### 6.1.6. Xanthan Gum

Xanthan gum (Figure 22) is a non-toxic polysaccharide secreted by the bacterium Xanthomas campestris via aerobic fermentation. Xanthan gum contains a repeating pentasaccharide pattern consisting of one glucoronic acid unit, two mannose units, and two glucose units. It is used as a thickener, rheological agent, emulsifier, and stabiliser [114]. It is known for its excellent pseudoplasticity properties, rheological properties, and relative stability under alkaline and acidic conditions [115]. Chemically modified xanthan gum has good antioxidant activity [116]. Low molecular weight or oligosaccharide xanthan gum exhibits good free radical scavenging activity.

The solubility of xanthan gum in cold water and its ionic properties provide mild conditions for the formulation step [116]. In addition, the high thermal stability of xanthan gum is usually superior to most other biopolymers. Xanthan is a promising encapsulation option, particularly when used in combination with other natural gums or biopolymers, to boost the retention capacity of active components within the polymer matrix. In the encapsulation of essential thyme oil through emulsion preparation, xanthan gum reduces diffusivity and improves stability, thereby enhancing encapsulation performance, unlike guar gum [117]. This water-soluble colloid provides an alternative wall material for micro- or nanoencapsulation of active ingredients such as acerola and ciriguela. Encapsulation can be carried out by spray drying. The micro- or nanobeads can also be stabilised by ionic cross-linking. The xanthan gum capsules showed an excellent protective effect on epithelial cells, preventing damage by hydrogen peroxide [116].

#### 6.1.7. Gum Arabic

Gum arabic (Figure 23) is as an excellent polymer for encapsulating cells and bacteria due to its protective properties. Extracted from acacia trees, gum arabic is only produced in regions such as Sudan and Nigeria. Gum arabic has some disadvantages, such as high cost and limited supply. It is widely used as a stabiliser in the food, pharmaceutical, adhesive, printing, textile, and paint industries. This polymer is a complex mixture of glycoproteins and polysaccharide. The gum mixture consists of three carbohydrates: arabinogalactan (88%), glycoprotein (2%) and arabinogalactan protein complexes [118]. In its native form, the highly branched structure is composed of high molecular weight macromolecules [11,56]. Above pH 2.2, the structure of this water-soluble gum is negatively charged due to carboxyl moieties [119,120].

This amphiphilic polymer can successfully act as an emulsifier and has good thermal stability. Gum arabic has been used as a stabiliser for oil-in-water emulsions [119]. The physicochemical properties, such as stability over a wide pH range and high oxidative stability, are of particular interest for the encapsulation of flavour compounds. The stabilisation of nano- and microbeads can be achieved by ionic cross-linking with calcium ions. A combination of gum arabic and alginate was used for the encapsulation of Enterococcus durans IW3. Bacteria viability was improved by the addition of gum arabic to the polymer matrix [121].

#### 6.1.8. Guar Gum

Guar gum (Figure 24) is a non-ionic galactomannan [23], with the main backbone consisting of β-D-mannopyranosyl units, to which side chains of α-D-mannopyranosyl units are attached. It is extracted from the seeds of Cyamopsis tetrago, a leguminous plant. This gum is used as a stabiliser and a thickener in the pharmaceutical, food [122], paper, and explosives industries [56].

Temperature affects the degree of hydration and the dissolution of guar gum in water solution. This biocompatible and biodegradable polymer is strongly hydrophilic and forms shear-thinning systems upon hydration without heating [123]. The polysaccharide is stable over a wide range of pH between 1 and 1. Guar gum solutions undergo a viscosity change even at very low concentrations, due to the interaction of galactose chains with water molecules and polymer entanglements. The recommended concentration is less than 1% [122]. The addition of sugar to guar gum mixtures can be used to reduce the viscosity of the medium. The sugar molecules compete with the water molecules, delaying the hydration of the guar gum [124]. Micro- or nanoencapsulation with guar gum has already been carried out using spray drying, e.g., encapsulation of *Rhibozium leguminasorum* bv. *Trifolii* [125], and oil-in-water emulsion methods, e.g., encapsulation of *astaxanthin* via Pickering formulation stabilised with chitosan/guar gum nanoparticles of guar gum/chitosan nanoparticles [126]. One of the disadvantages of this non-ionic biopolymer is the use of additional cross-linkers to stabilise the polymer network, such as glutaraldehyde [56]. Guar gum has been used as the drug delivery matrix mainly because of its interaction with mucin, which makes it useful in the development of mucoadhesive formulations. Poly(lactic-co-glycolic acid) nanoparticles have been combined with mucoadhesive guar gum films for the delivery of anti-hypersensitive peptides [127].

#### 6.1.9. Agarose

Extracted from the wall cells of the red algae Rhodophyta, agar–agar is composed of natural polysaccharides. The major neutral polysaccharide is agarose (Figure 25), with agarobiose as the repeating unit. Agar–agar is also composed of charged agaropectin [128]. Agaropectin is a polysulphated polysaccharide, and can also be substituted by methoxylate, pyruvate, and gluconate residues.

The ability to gel at different temperatures makes it more valuable for formulation. Agarose is insoluble in cold water but dissolves easily in boiling water. On cooling, the gelation temperature of agarose gel is observed to be between 30 and 45 °C. The formation of the 3D network is promoted by hydrogen bonding. The porosity of the structure is regulated by the agarose concentration [129]. The encapsulation efficiency of poorly water soluble drugs is compromised by the difficulty of entrapping hydrophobic compounds into the polymer matrix [130]. Agarose nanoparticles have been prepared for the delivery of enzymes and proteins. Indeed, the ability to catalyse the oxidation of the substrate 2,2′-azinobis (3-ethylbenzthiazoline-6-sulphonate) in the presence of hydrogen peroxide by the enzyme horseradish peroxidase was enhanced after encapsulation of the enzyme in the agarose matrix. Encapsulation increases the affinity of the enzyme for its substrate. The higher the concentration of agarose, the higher the reaction rate. The rates were 3.1 µM·s^−1^ for 0.5% agarose, against 5.5 µM·s^−1^ for 2% agarose. The study showed that the higher the degree of entrapment, the higher the stability of the proteins. Protein stability was increased by the degree of entrapment due to the preservation of the enzymatic conformation and the shielding effect of the porous structure [131].

#### 6.1.10. Dextrin: Maltodextrin and Cyclodextrin

Dextrins are polysaccharides used as additives in food processing. Dextrins are produced by hydrolysis of the starch chain. These biodegradable and biocompatible biopolymers are freely soluble in water and slightly soluble in anhydrous alcohol. Maltodextrins are classified according to their dextrose equivalent (DE). The DE ranges from 3 to 20. The higher the DE, the shorter the glucose chain. Based on their structure, dextrins can be divided into maltodextrins, which are characterised by short linear chains, and cyclodextrins, which are characterised by a circular structure (Figure 26). The advantages of using dextrins as encapsulants are their low cost and their ability to protect molecules such as flavourings from oxidising conditions. Hydrolysed starches have been reported to improve the shelf life of orange oil and carotene [132].

Dextrins are often used in combination with other encapsulants, such as gums and starches. Maltodextrins have been used with whey proteins to encapsulate lime essential oil. Particles containing only maltodextrins form a denser and more oxygen-permeable wall system, which has been shown to improve the storage stability of betalain biological pigment [23,133].

The degree of polymerisation (DP) of maltodextrin chains affected the entire encapsulation process, from encapsulation efficiency to the release and protection of the encapsulated lime essential oil, including the morphology of the beads (Figure 27). The microencapsulation of essential oils affects the stability during storage, the controlled release, and the degradation of the encapsulated bioactive compounds. The microcapsules made from whey proteins and high DP maltodextrins showed a better protective performance. The low DP maltodextrins tend to form a more porous polymer structure due to their molecular structure. The pores facilitate the diffusion of moisture and oxygen into the microparticles, accelerating the degradation reactions of bioactive compounds [134].

The most common types of cyclodextrins are α-, β- and γ-cyclodextrin, respectively composed of 6, 7, and 8 dextrin units [23]. Cyclodextrins have high stability under alkaline conditions and, due to their cylindrical structure, can be used as a polymer matrix for encapsulating active ingredients. Cyclodextrins are characterised by their hydrophilic circular truncated cone shape. When combined with a molecule, cyclodextrins can easily form a host-guest supramolecular complex [135]. This cone shape has a hydrophobic hollow conical cavity with a depth of 7.9 Å [136]. The cavity is suitable for the incorporation of hydrophobic guest molecules, depending on the size of the core part. The main drawbacks are the cost and sensitivity to acid hydrolysis at low pH.

Curcumin is a natural polyphenolic compound known for its interesting properties such as antioxidant, anti-inflammatory, antibacterial antiviral, anticancer, antidiabetic, and neuroprotective properties. Curcumin has been encapsulated in β-cyclodextrin to improve absorption of the active compound through the intestinal tract and to avoid rapid metabolism by the liver. The incorporation of curcumin compounds was performed using the solvent evaporation method due to its simplicity. This method provided a light-yellow, fluffy powder with excellent aqueous solubility. After the incorporation of curcumin, the complex was encapsulated in chitosan to ensure the transport to human skin cancer cells, which showed higher cytotoxicity [135].

#### 6.1.11. Locust Bean Gum

Locust bean gum (Figure 28) is a galactomannan extracted from the seeds of the locust bean tree with a non-ionic chemical structure similar to guar gum. This polysaccharide is used as a texturising agent in the food industry. Its solubility and rheological properties are mainly determined by the molecular conformation of the chains [137]. Locust bean gum has been co-formulated with κ-carrageenan to fabricate hard gel capsules [138].

Locust bean gum readily forms gels when combined with other hydrocolloids. Drug delivery systems using locust bean gum have shown promising results for non-toxic oral administration in the treatment of colorectal cancer. For example, a combination of sodium alginate and locust bean gum was used to create an interpenetrating polymeric network for the delivery of the anticancer drug Capecitabine via the ionotropic gelation method. This approach resulted in prolonged drug release (Figure 29) and improved bioavailability of the drug due to its encapsulation within the hydrogel structure [7].

#### 6.1.12. Chitosan

Chitosan, constituted of D-glucosamine and N-acetyl-D-glucosamine units linked by β-(1 → 4) glycosidic bonds [6], is a linear polysaccharide derived from the deacetylation of chitin via alkaline hydrolysis [139] or enzymatic treatment (Figure 30). Chitin deacetylase, a secreted enzyme, catalyses the hydrolysis of acetamido groups in chitin, forming the polycationic polymer chitosan [140]. Innovative deacetylation methods using deep eutectic solvents composed of glycerol, potassium carbonate, and choline acetic acid have been demonstrated as promising and environmentally friendly conditions [141]. The molecular weight, the degree of deacetylation (reflecting the content of acetylated and free-amino groups in chitosan), and the polydispersity of polymer chains significantly influence its physical, biological, and chemical properties, including viscosity and solubility [139,142]. Chitosan is soluble in neutral and acidic media; however, effective dissolution requires pH adjustment. The efficiency of chitosan dissolution using hydrochloric and acetic acids has been evaluated [143]. Chitosan’s chemical characteristics, particularly its positively charged functional groups, impart anti-inflammatory, antibacterial, and hemostatic properties. Its antibacterial efficacy increases with higher molecular weight and degree of acetylation, significantly reducing populations of bacteria such as *Escherichia coli* [144,145].

In drug delivery applications, chitosan has been used as a wall material. Chitosan-alginate nanoparticles have been developed for the stable encapsulation of vitamin B2 for oral administration, demonstrating potential as an encapsulating system in food matrices [146]. This system shows stability for approximately five months due to ionic cross-linking, Van der Waals forces, and hydrogen bonding between the biopolymers and vitamin B2. In acidic conditions (pH below the pKa of chitosan around 6.5 and above the pKa of alginate around 4), chitosan forms an ionic cross-linked network with negatively charged polymers, inducing gel formation [146,147].

Chitosan is also used in drug delivery systems targeting mucosal surfaces due to its positive charge surface. In the development of medical devices for localised vaginal therapy, chitosan-coated liposomes exhibit superior binding efficiency with porcine mucin (around 75%) compared to plain liposomes (around 50%) [148].

### 6.2. Proteins

Proteins serve as coating materials for the encapsulation of active ingredients due to their unique functional properties that facilitate gel formation. Made up of amino acids arranged in repeating units, proteins act as polyelectrolytes, with the overall charge of their chains determined by their isoelectric point. This is the specific pH at which proteins carry a neutral charge. Therefore, by adjusting the pH of the surrounding medium, the total charge of the proteins can be changed. The isoelectric point is a critical parameter for optimising formulation and microbead formation when working with proteins. They can also be mixed with polysaccharides to create complex encapsulation systems [85,107].

#### 6.2.1. Gelatin

Gelatin (Figure 31) is a protein of animal origin obtained from the chemical degradation of collagen by partial hydrolysis. This inexpensive and natural raw material is extracted from the skin of cold fish, but also from bovine bones, hides, and pig skins [149].

With a high molecular weight ranging from 65,000 g·mol^−1^ to 300,000 g·mol^−1^, its structure is characterised by 18 different types of amino acids, of which glycine, proline, and hydroxyproline are the most abundant. Used as texturiser and gelling agent, gelatin forms a high-viscosity solution in water at 40 °C, which gels on cooling [107]. The structure of gelatin can be modified enzymatically and chemically to optimise hydrogel formation and alter the final properties of the gel. The strength of the hydrogel network can also be tailored by the use of cross-linking agents such as glutaraldehyde. Gelatin is often chosen as a carrier material due to its elastic and robust properties. In addition, its amphoteric nature gives it the potential ability to interact with anionic polysaccharides such as gellan gum [107]. Due to its positive charge below its isoelectric point, which is typically around pH 5, gelatin is often paired with gum arabic. This combination is favoured by their opposite charges at low pH levels. When exposed to electrostatic interaction and low temperatures, they undergo a process that triggers the formation of insoluble particles. It is also a suitable polymer for the encapsulation of hydrophilic and hydrophobic substances, such as vegetable oils or oil-soluble dyes [23]. For example, the encapsulation of the hydrophobic lycopene colorant can be achieved through spray drying using a wall system composed of gelatin and sucrose [150]. Microencapsulation with sodium alginate and gelatin has been used to shield and improve the thermal stability of paraffin [151].

#### 6.2.2. Milk Proteins: Whey Protein and Casein

Milk proteins, consisting of whey proteins and casein (Figure 32) with a ratio of 20% and 80%, respectively in cow’s milk [152], are cost-effective biopolymers with many functional and structural properties. These proteins are low-cost biopolymers with many functional and structural properties. With an isoelectric point of around 4.5, these pH-sensitive compounds are highly suitable as key components in the formulation of “vehicles” designed to deliver various bioactive compounds [23,153,154].

Whey proteins are mainly composed of β-lactoglobulin, a small globular protein. Acting as a natural protective barrier, they are used in various functional food products, acting as emulsifiers, gelling agents, foam stabilisers, water binders, and film formers [85]. A study on the encapsulation of curcumin highlights the improved solubility and bioavailability of curcumin when microencapsulated with whey proteins using the spray drying technique [155]. Whey proteins have also been co-formulated to improve microencapsulation properties. They have been combined with alginate to produce uniform microcapsules with improved gastrointestinal resistance. By exploiting the interaction between whey protein and alginate within a pH range of 2 to 4, capsules with small pores have been designed to encapsulate *L. acidophilus*, providing protection for the bacteria [85].

Caseins are characterised as proline-rich, open-structured proteins with distinct hydrophobic and hydrophilic domains. Due to their structure, caseins have a natural tendency to self-assemble into spherical structures called micelles, which typically range in diameter from 50 to 500 nm [156]. Due to their amphiphilic nature, caseins can encapsulate hydrophobic compounds in a polar solvent, forming direct micelles, and hydrophilic compounds in a non-polar solvent, forming reverse micelles [157]. Caseins are particularly advantageous for encapsulating hydrophobic compounds due to the hydrophobic core of casein micelles. However, controlling the dissociation and association mechanisms of the micellar structure is a challenge in this encapsulation method. Studies have investigated the stability of micelles and identified disruption of calcium bridges and hydrophobic interactions as causes of micellar dissociation. Factors such as temperature, pH, salt concentration, presence of organic solvents, and chelating agents can significantly affect the functional stability of casein micelles [158].

#### 6.2.3. Soy and Pea Proteins: Legume-Based Proteins

Soy and pea proteins are the main legume proteins used for encapsulation and are particularly valued for their nutritional properties in the context of encapsulation for the food industry. The advantages of food proteins lie in their chemical and structural versatility [159]. These proteins provide a cost-effective alternative for shielding and ensuring the delivery of bioactive compounds and are used to encapsulate both hydrophilic and hydrophobic bioactive compounds [160]. Encapsulation with legume-based proteins can be carried out by spray drying [161], coacervation [162], and extrusion methods [1].

However, sensitivity to pH, light and thermal treatments, and hydrophobicity or low water solubility may limit their use [159]. Pea proteins are obtained from yellow peas, while soy proteins are obtained from soy beans. Above the isoelectric point pH of about 4.5, the soy isolate carries a negative charge. In a study highlighting the ability of proteins to bind and form complexes with blueberry polyphenols, pea proteins showed the highest encapsulation efficiency by spray drying (Figure 33) and were found to be the most effective in stabilising phytochemicals extracted from wild blueberry pomace compared to wheat, chickpea, and coconut flour [163].

## 7. Conclusions

This review provides an overview of research on encapsulation in organic polymers. The theoretical background presented helps to understand the polymer behaviour throughout the encapsulation process. Various biopolymers are presented along with their physicochemical properties to highlight their advantages and drawbacks as coating materials. Common and cost-effective encapsulation techniques such as extrusion, spray-drying, and coacervation are discussed.

The aim of this review is to stimulate the exploration of new alternatives based on existing research. There is a clear need for continued research into encapsulation to better meet economic, environmental, and technical requirements. Encapsulation is a promising technology that provides both protection and precise control over the release of active ingredients. Its application spans a wide range of fields, encouraging innovation and the development of new bioprotective tools, for example.

## Figures and Tables

**Figure 1 materials-17-02774-f001:**
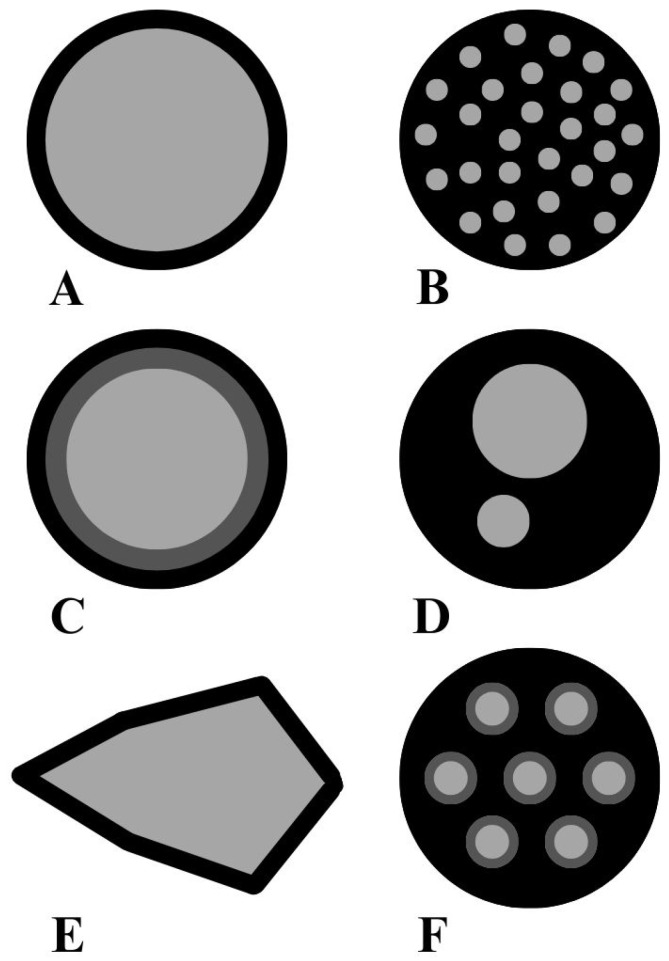
Different types of microcapsule architectures: (**A**) simple microcapsule, (**B**) microsphere, (**C**) multiwall microcapsule, (**D**) multicore microcapsule, (**E**) irregular microcapsule, (**F**) assembly of microcapsules, polymer layer are represented in black and dark grey and the light grey represents the active substance, inspired by [20].

**Figure 2 materials-17-02774-f002:**
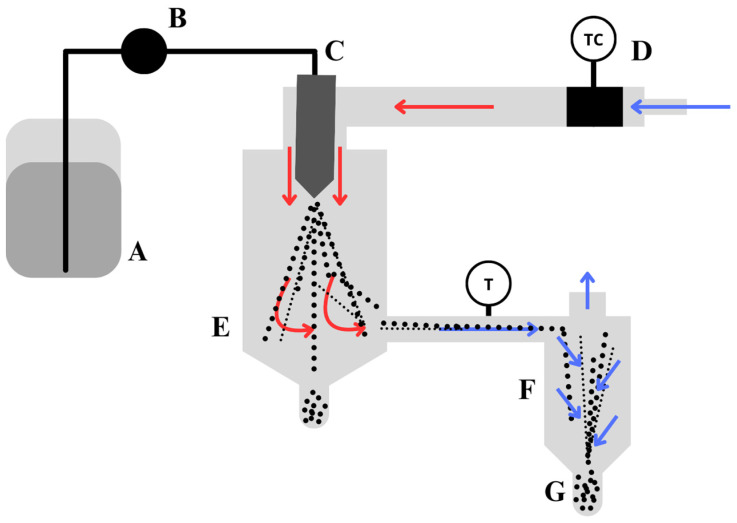
Scheme of the spray-drying process for encapsulation of active ingredients, (**A**) tank containing the sprayed mixture: polymer and active ingredient, (**B**) pump to feed the mixture in the system, (**C**) spray nozzle, (**D**) heater to heat up the airflow, (**E**) chamber, (**F**) cyclone separator, and (**G**) spray dried capsules, inspired by [4].

**Figure 3 materials-17-02774-f003:**
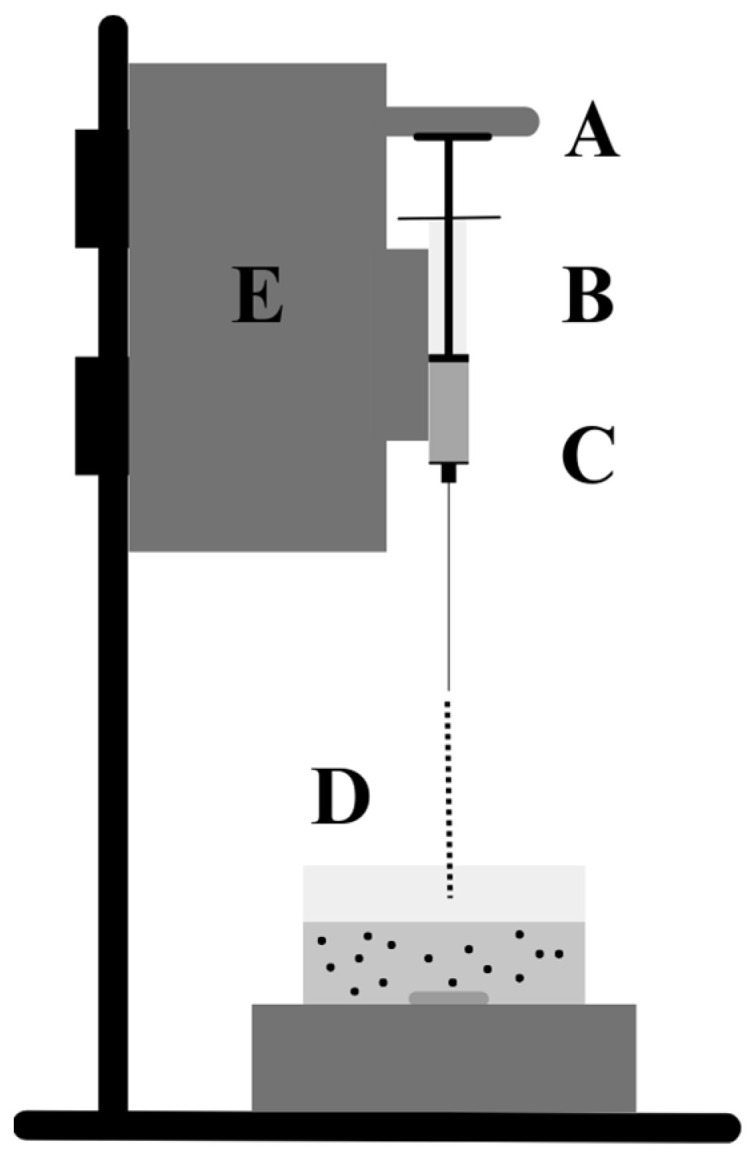
Scheme of the extrusion process, (**A**) Syringe, (**B**) Polymer (alginate) solution, (**C**) Syringe needle, (**D**) Gelling bath (with calcium chloride), and (**E**) Syringe pump, inspired by [11].

**Figure 4 materials-17-02774-f004:**
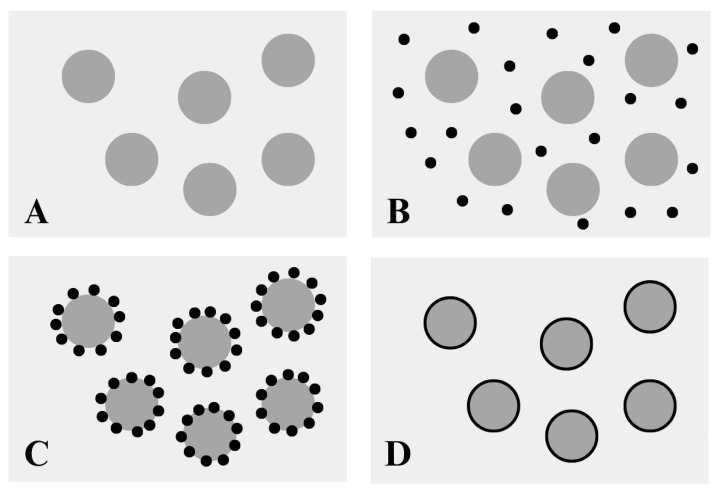
Mechanism of microencapsulation formation by the coacervation method, (**A**) suspension of the core material (dark grey circles) in the liquid phase (light grey background), (**B**) suspension of the polymer (small black circles) in the liquid phase, (**C**) adsorption of the polymer material onto the core material, and (**D**) gelation and solidification of the microcapsule wall (black layer surrounding the dark grey circles), inspired by [15].

**Figure 5 materials-17-02774-f005:**
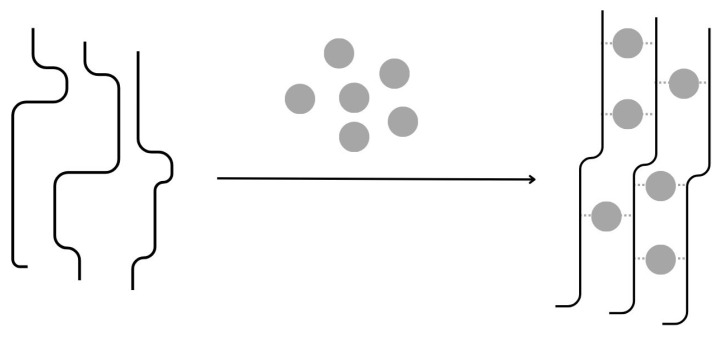
Mechanism of gum-based micro- and nanoparticle formations: Ionotropic gelation, inspired by [56].

**Figure 6 materials-17-02774-f006:**
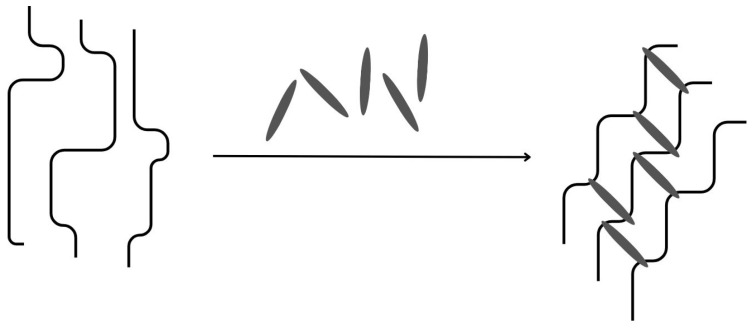
Mechanism of gum-based micro- and nanoparticle formations: Covalent cross-linking, inspired by [56].

**Figure 7 materials-17-02774-f007:**
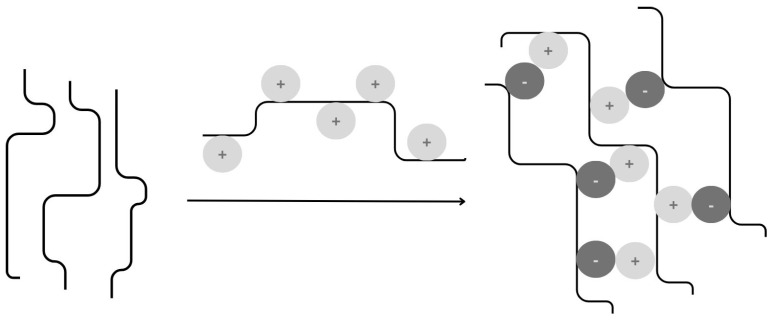
Mechanism of gum-based micro- and nanoparticle formations: Polyelectrolyte complexation, inspired by [56].

**Figure 8 materials-17-02774-f008:**
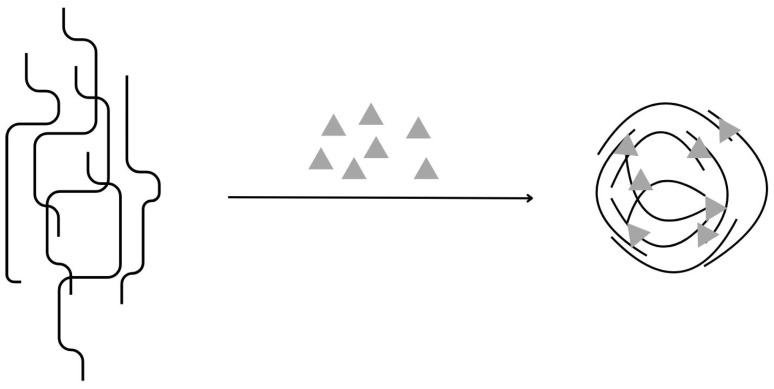
Mechanism of gum-based micro- and nanoparticle formations: Drug or hydrophobic agent/polymer conjugation with self-assembly, inspired by [56].

**Figure 9 materials-17-02774-f009:**
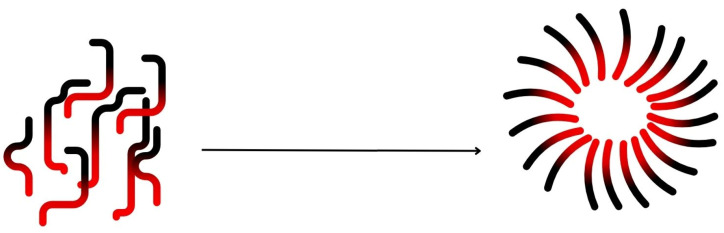
Mechanism of gum-based micro- and nanoparticle formations: Self-assembly of amphoteric molecular compounds, inspired by [56].

**Figure 10 materials-17-02774-f010:**
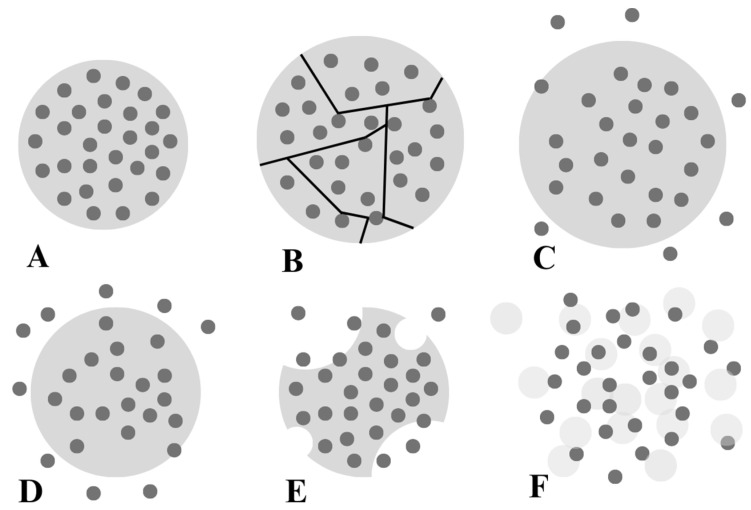
Mechanisms of the release of active substances, from polymer coatings, (**A**) initial encapsulation system, (**B**) fragmentation, (**C**) swelling, (**D**) diffusion, (**E**) degradation, and (**F**) dissolution, inspired by [4].

**Figure 11 materials-17-02774-f011:**
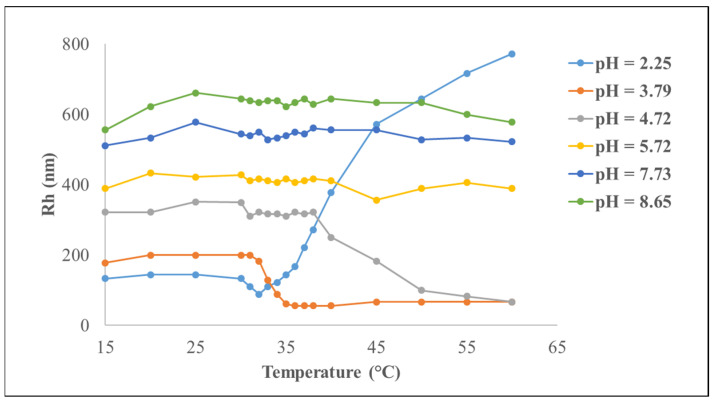
Plot of the hydrodynamic radius of poly (N-isopropylacrylamide-co-acrylic acid) polymer microgel according to the temperature at different pH values. Reproduced with permission from Farooqi et al., Arab. J. Chem.; published by Elsevier, 2017 [83].

**Figure 12 materials-17-02774-f012:**
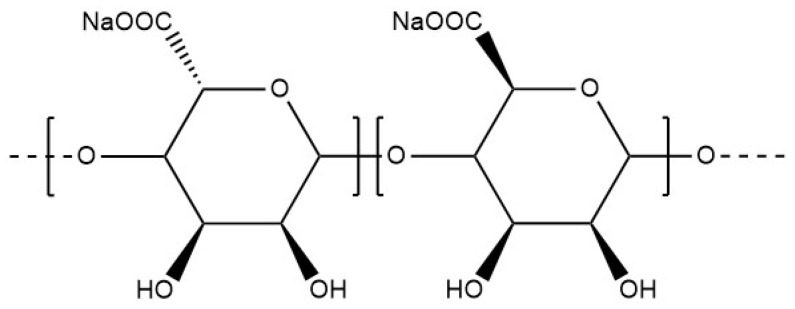
Molecular structure of sodium alginate.

**Figure 13 materials-17-02774-f013:**
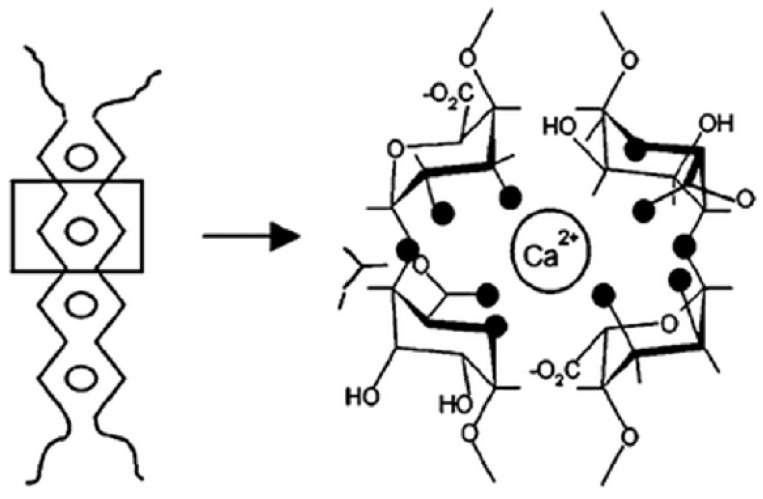
“Egg box” model for calcium alginate, Reproduced with permission from Finotelli et al., Polimeros; published by ABPol, 2017 [90].

**Figure 14 materials-17-02774-f014:**
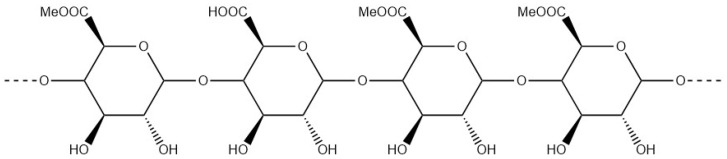
Molecular structure of pectin: homogalacturonan structure.

**Figure 15 materials-17-02774-f015:**
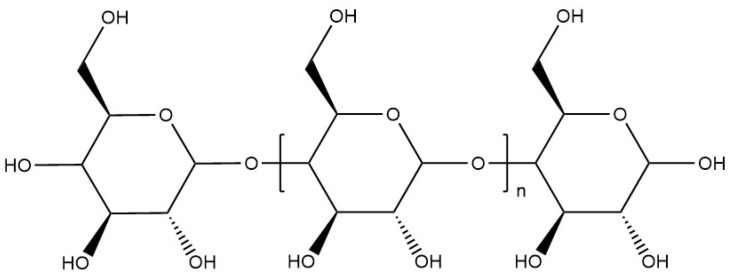
Molecular structure of amylose.

**Figure 16 materials-17-02774-f016:**
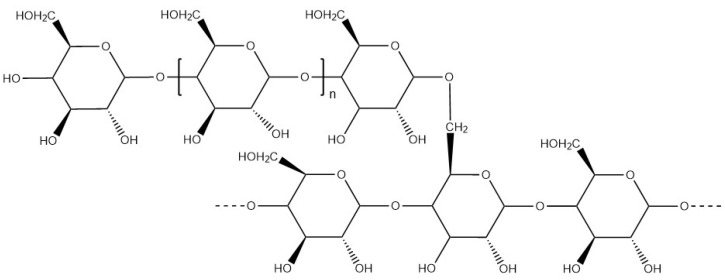
Molecular structure of amylopectin.

**Figure 17 materials-17-02774-f017:**
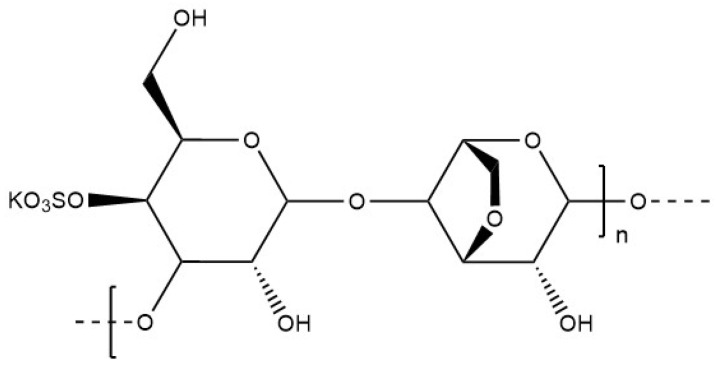
Molecular structure of κ-carrageenan.

**Figure 18 materials-17-02774-f018:**
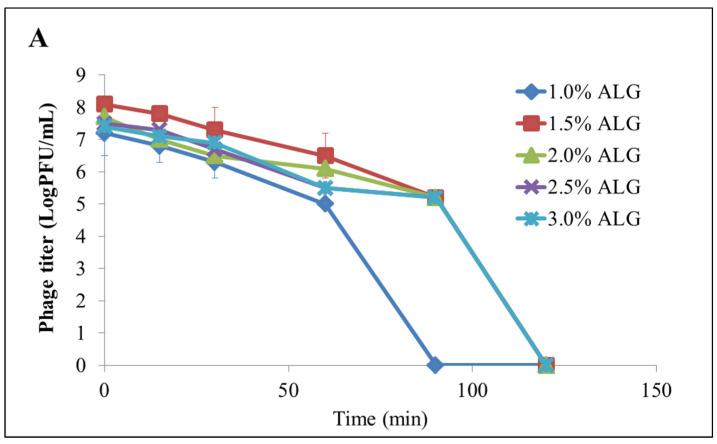
Stability and release characteristics of release of phage encapsulated inside microcapsules by in vitro digestion, (**A**) pure ALG microcapsules. (**B**) AC microcapsules AC1–AC9 represent polysaccharide mixtures in different proportions. (AC1) 1%ALG and 0.15%CG, (AC2) 1.5%ALG and 0.15%CG, (AC3) 2%ALG and 0.15%CG, (AC4) 1%ALG and 0.3%CG, (AC5) 1.5%ALG and 0.3%CG, (AC6) 2%ALG and 0.3%CG, (AC7) 1%ALG and 0.45%CG, (AC8) 1.5%ALG and 0.45%CG, (AC9) 2%ALG, and 0.45%CG. Reproduced with permission from Zhou et al., Front. Microbiol.; published by Frontiers, 2022 [109].

**Figure 19 materials-17-02774-f019:**
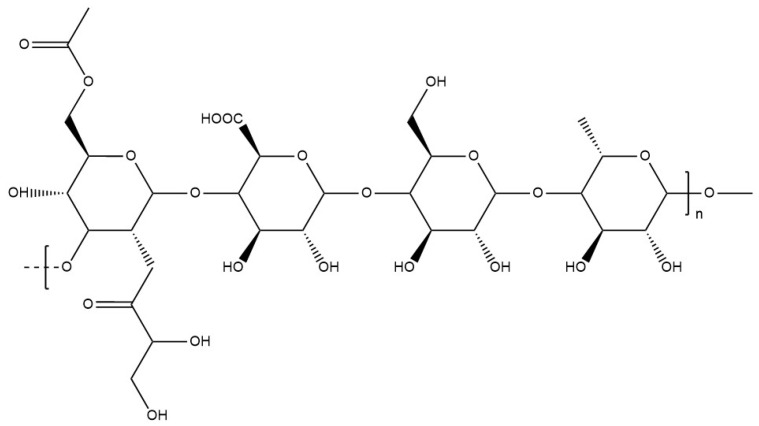
Molecular structure of high acyl gellan.

**Figure 20 materials-17-02774-f020:**
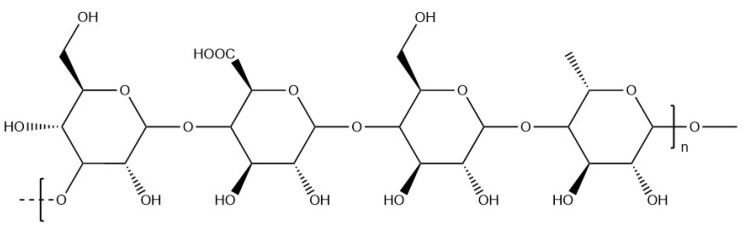
Molecular structure of low acyl gellan.

**Figure 21 materials-17-02774-f021:**
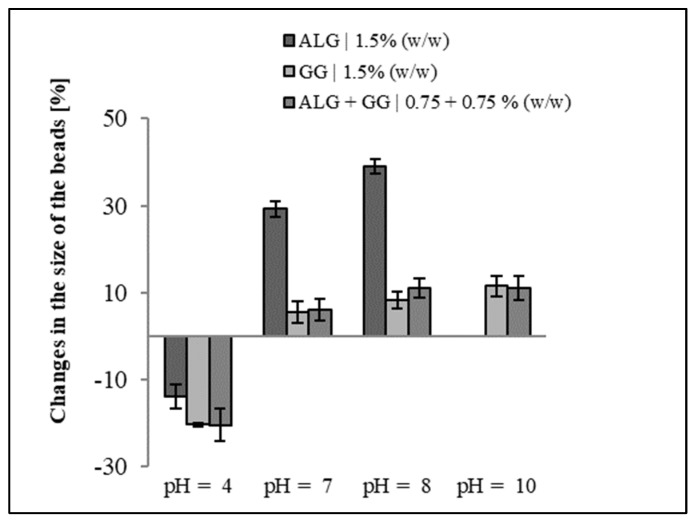
Percentage change in bead size (%) after 24 h of immersion in different pH solutions. Values are shown with the standard deviation. Adapted from [40].

**Figure 22 materials-17-02774-f022:**
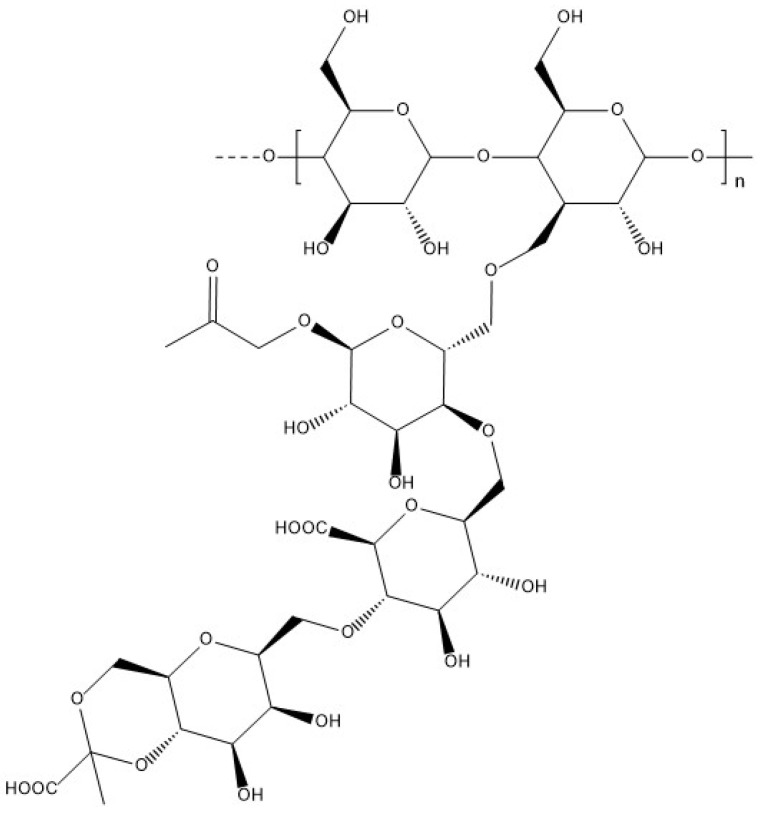
Molecular structure of xanthan gum.

**Figure 23 materials-17-02774-f023:**
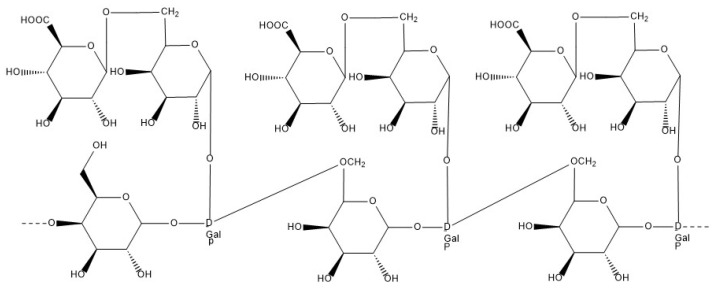
Molecular structure of gum arabic.

**Figure 24 materials-17-02774-f024:**
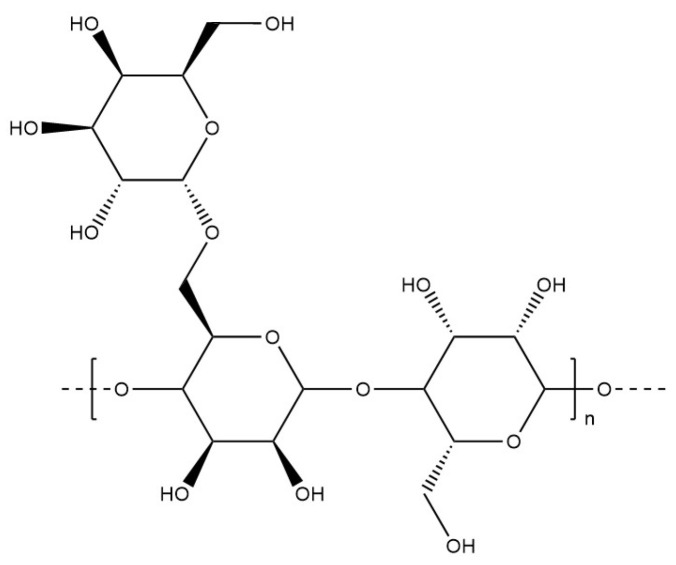
Molecular structure of guar gum.

**Figure 25 materials-17-02774-f025:**
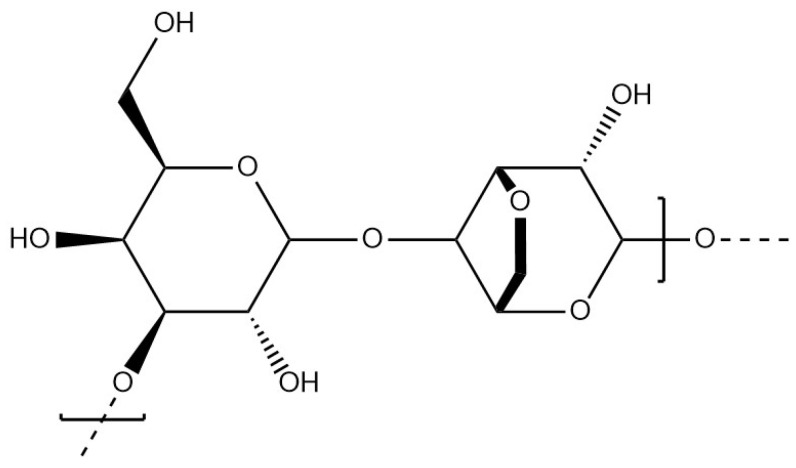
Molecular structure of agarose.

**Figure 26 materials-17-02774-f026:**
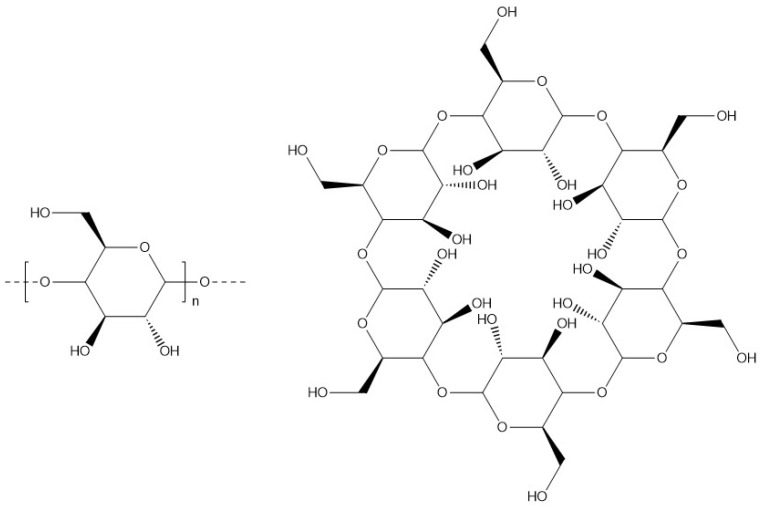
Maltodextrin (**left**) and cyclodextrin (**right**) molecular structures.

**Figure 27 materials-17-02774-f027:**
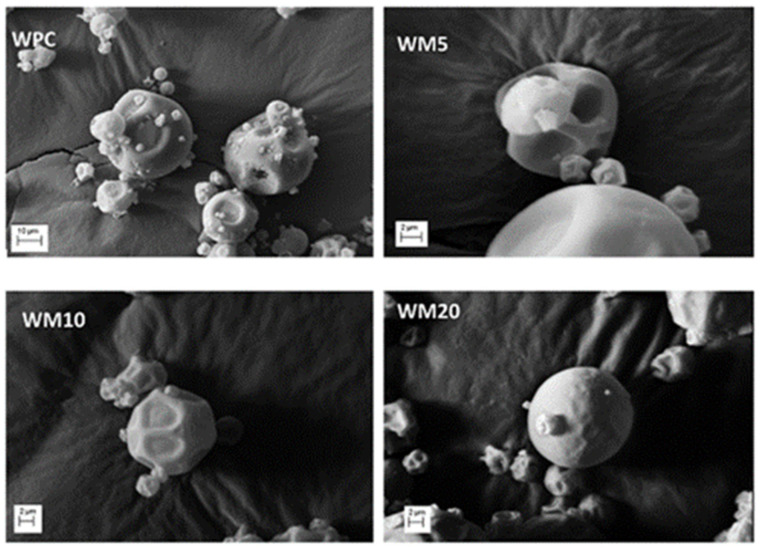
Morphology of lime essential oil microparticles: Encapsulation by spray drying method of lime essential oils in whey protein concentrate, whey protein blended/maltodextrin DE5 (WM5), whey protein blended/maltodextrin DE10 (WM10), and whey protein blended/maltodextrin DE10 (WM20). Reproduced with permission from Campello et al., Food Res. Int.; published by Elsevier, 2018 [134].

**Figure 28 materials-17-02774-f028:**
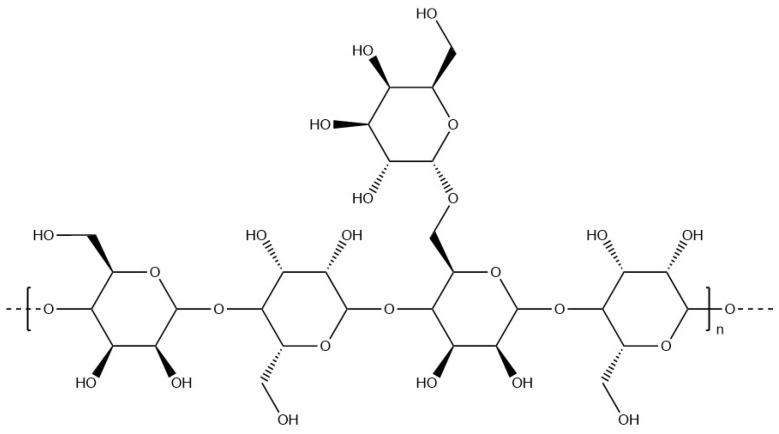
Molecular structure of locust bean gum.

**Figure 29 materials-17-02774-f029:**
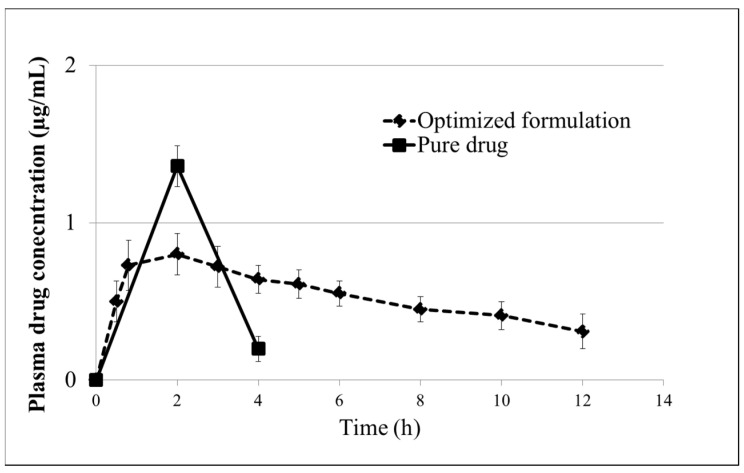
Characterisation of the controlled release of Capecitabine in vitro, plasma drug concentration versus time profile of the Capecitabine encapsulated in locust bean gum/alginate microbeads, vertical bars represent mean ± S.D. (standard deviation), the total number of values is 6. Reproduced with permission from Upadhyay et al. Mater. Sci. Eng. C; published by Elsevier, 2019 [7].

**Figure 30 materials-17-02774-f030:**
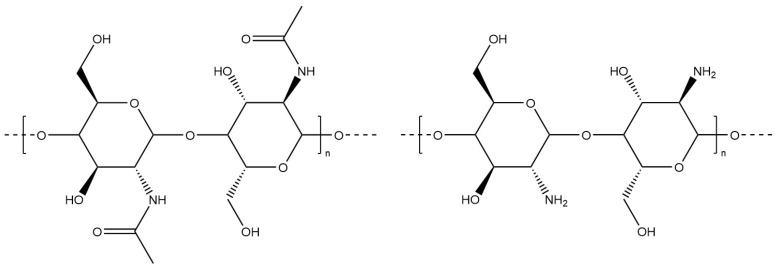
Molecular structure of chitin and chitosan.

**Figure 31 materials-17-02774-f031:**
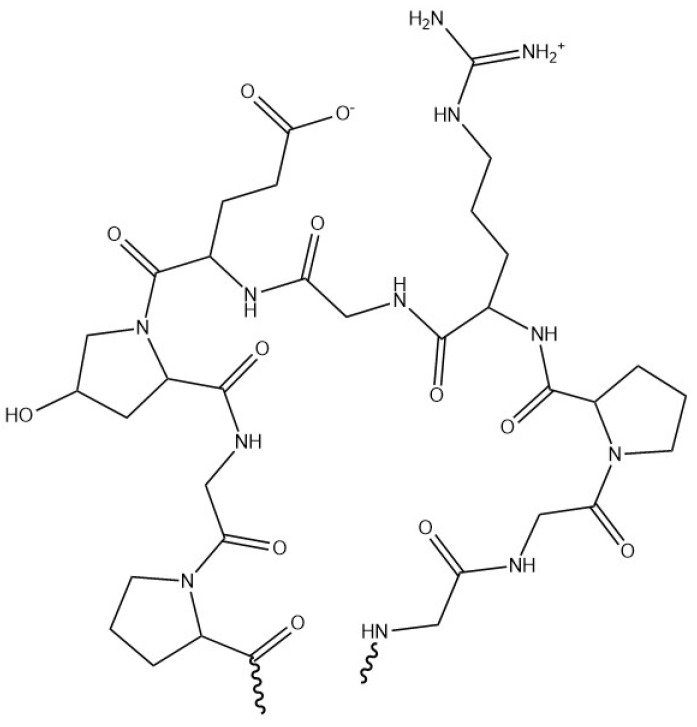
Molecular structure of gelatin.

**Figure 32 materials-17-02774-f032:**
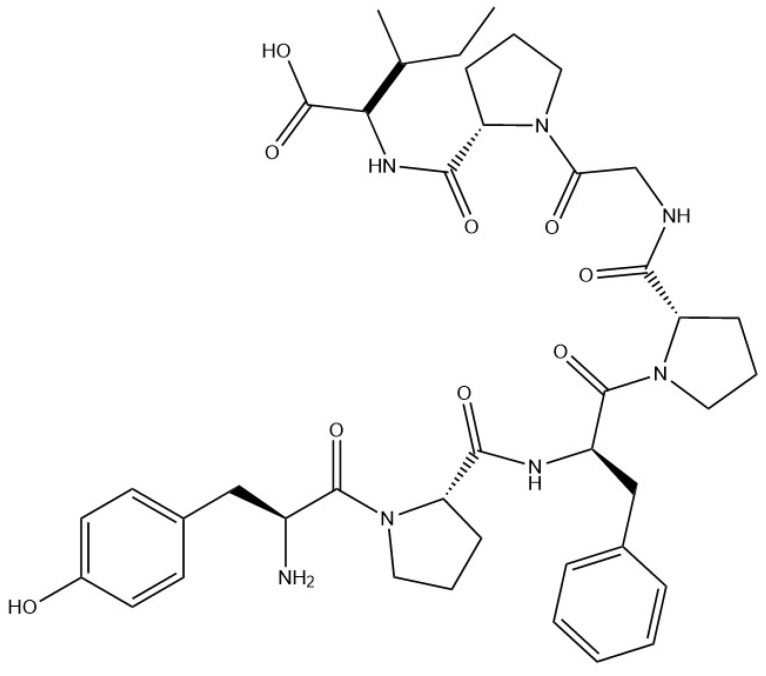
Molecular structure of casein.

**Figure 33 materials-17-02774-f033:**
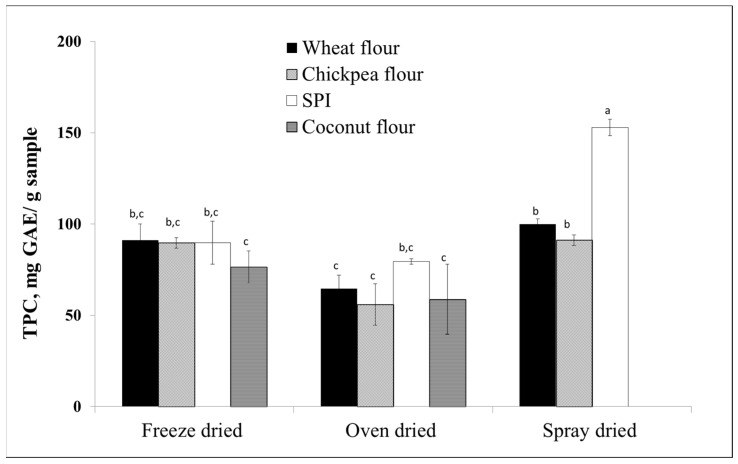
Total phenolic content of blueberry polyphenol-protein matrices, the scavenging capacity of different combinations of blueberry polyphenol-protein matrices obtained by different entrapment methods: Freeze drying, oven drying, and spray drying (total phenolic content (TPC) calculated as mg gallic acid equivalent). Bars with different letters (a,b,c) are significantly different by Tukey’s test (2-way ANOVA) test, *p* < 0.01. Reproduced with permission from Correia et al. Food Chem.; published by Elsevier, 2017 [163].

**Table 1 materials-17-02774-t001:** Overview of the encapsulation methods.

**Physical Methods**		
**Name**	**Description**	**Example**
Spray drying	cf. part 2.1	cf. part 2.1
Extrusion	cf. part 2.2	cf. part 2.2
Freeze drying	Involve freezing the solution containing the active substance and then reducing the surrounding pressure to allow the frozen solvent to sublimate, leaving behind a porous structure with encapsulated materials.	Encapsulation of *Elsholtazia ciliata* ethanolic extract using various coating materials [27].
Spray cooling	Atomisation of a mixture of the active substance and a liquefied lipid carrier at low temperatures, using a cooling medium to solidify the particles [28].	Microencapsulation of heat sensitive compounds such as vitamine B12 [29].
Electrospinning	Use of high-voltage electric field to produce fibres of polymers encapsulating active ingredients within the fibre matrix.	Fragrance encapsulation in polyvinylalcohol matrix by emulsion electrospinning [30].
Electrospray	Use of a high-voltage field to create droplets from liquid solution, which then solidify to form particles.	Manufacture of poly(lactic acid) nanoparticles incorporating antithrombotic drug [31].
Fluidised bed coating	A coating solution is sprayed onto particles suspended in an upward air flow. As the solvent evaporates, coated particles are left behind.	Assessing the encapsulation of orange oil: a comparison between spray drying/agglomeration and fluidised bed granulation [32].
**Chemical Methods**		
**Name**	**Description**	**Example**
Polymerisation	The encapsulation of active ingredients within the resulting polymer matrix occurs through the in situ polymerisation of monomers.	Encapsulation of the chlorinated flame retardant with poly(methyl methacrylate) and poly(1-vinyl-2-pyrrolidone) through in-situ dispersion polymerisation in supercritical carbon dioxide [33].
Interfacial polymerisation	Polymerisation takes place at the interface of two immiscible phases, resulting in the formation of a polymer shell around the dispersed droplets containing the active ingredient.	The microencapsulation of *Cypermethrin* is achieved through the interfacial polymerisation of polyuria [34].
**Physicochemical Methods**		
**Method**	**Description**	**Example**
Double emulsion	A water-in-oil-in-water (W/O/W) or oil-in-water-in-oil (O/W/O) emulsion is formed to encapsulate hydrophilic or lipophilic ingredients within the inner phase, allowing control over the capsule architecture.	Development of a microfluidic synthesis method for advanced microparticles involves encapsulating dye within water-oil-water droplets stabilised by diblock polymers [35].
Emulsification	Encapsulation via the formation of an emulsion (typically oil-in-water or water-in-oil), where the active ingredient is dispersed in one phase and the polymer forms a shell around the droplets upon solvent removal.	Manufacture of delivery systems of polyphenols via formulation of oil in water emulsion [36].
Microemulsion	Thermodynamically stable emulsion with droplet size in the nanometer range, used for the encapsulation of active ingredients within the dispersed phase.	Encapsulation of the antibiotic *Levofloxacin* in a biocompatible microemulsion involving clove oil stabilised in water with tween 20 as a surfactant and 2-propanol as a co-surfactant [37].
Coacervation	Simple coacervation is a physical approach Complex coacervation chemical is a chemical approach.	cf. part 2.3
Solvent in chemical reaction	Use of solvents in reactions to facilitate encapsulation through methods such as solvent evaporation, nanoprecipitation, or other reaction-based encapsulation techniques.	Synthesis of Poly(lactic-coglycolic acid) microspheres prepared by evaporation of dichloromethane and dimethylsulfoxide [38].
Layer-by-layer	Sequential deposition of alternating layers of oppositely charged polymers onto a core material, creating a multilayered shell around the core.	Encapsulation of probiotics for delivery to the microbiome by deposition of successive layers of chitosan and alginate around bacteria [39].

## Data Availability

No new data were created in the framework of this review.

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
