# Peer review of "Encapsulation of Active Substances in Natural Polymer Coatings"

_materials, 2024, doi:10.3390/ma17112774_

Round 1
Reviewer 1 Report
Comments and Suggestions for Authors
The review discusses the encapsulation of active ingredients into structures produced by natural polymers. The topic is interesting but the manuscript is too short. Much more information is available and is required for this topic References are completely inadequate. Among them, there are many reviews. This is not bad by itself. A large number of specific cases and examples need to be reported as well. Moreover, the authors continuously refer to some of these reviews in separate cases leaving the eventual reader to search each time the relevant paragraph or chapter to locate the initial paper. This renders the work non-informative
The quality of the majority of figures is deficient to unacceptable. There is also inconsistency: some of the chemical formulas are acceptable and others are produced most probably by different software and are of very low quality. The structure of the review is also problematic. I do not understand why the mechanisms of the active compound release are discussed before the encapsulation technologies. Overall, in my opinion, there is considerable sloppiness in this work and it is not up to the standards of “Materials”
Comments on the Quality of English LanguageThe English language is OK
Reviewer 2 Report
Comments and Suggestions for Authors
Although the review of the active substance encapsulation by Akpo and colleagues contains considerable information, the current version is unsuitable for publication. The manuscript contains redundant information and repetitions. Many figures are blurry, and many chemical schemes are of poor quality. These critical weaknesses need thorough revision and the oversight of a native English-speaking chemist before resubmission.
The following issues require corrections without providing an exhaustive list. Only a few contextual redundancies and repetitions are listed below because the reviewer stopped to flag them all at some point to avoid a lengthy referee's report.
- The first two sentences of the abstract need reconsideration because of redundancies.
- The reviewer would appreciate an explanation of the repetition of "encapsulation" and "natural coating" in the Keywords section. These are included in the title and do not add any value to the indexing of the article.
- Lines 67-74 contain repetitions. So do lines 99-100, 108-111, etc. Please read the full text carefully and correct any repetitions.
- Figure 1 shows an irregular microcapsule (Figure 1C), but Chapter 2 begins with "Microcapsules are ... spherical systems ...". Is it only the reviewer who feels the contradiction? On the other hand, the "and" connection between semipermeability and sphericity is untrue.
- The referee assumes that usually, the swelling is before or parallel with the fragmentation. Drawing simultaneous processes is challenging, but the authors can try it.
- The logical link between Figure 2 and the subchapters is missing.
- Figs. 3, 14, 25, and 28 are blurred, and their higher-quality reconstruction is mandatory.
- As many schemes are horrible, the referee would recommend using a chemical drawing program. Because there are many free alternatives to ChemDraw (e.g., ACDLabs Sketch, Biovia Draw, etc., or several online applications), re-drawing the structures does not bring additional costs to the authors.
- Fig. 8 is about an alginate, but the alginates are later. Please relocate the figure.
- Although the number of references is commendable (but its number is less than 60), it may not be sufficient for a comprehensive analysis of such a large area.
Comments on the Quality of English LanguageAlthough there are no major linguistic flaws, due to the numerous repetitions and lengthy sentences of similar meaning, a native English control appears necessary.
Reviewer 3 Report
Comments and Suggestions for Authors
I would like to recommend this manuscript for publication after minor revision:
1. What's the figure in Line 484?
2. The authors have listed 20 figures in this review. Are they all representative or necessary?
3. The authors have listed a large number of molecular structure diagrams in this review. Can they be integrated into a large figure for comparison, or can they be integrated with their corresponding characterization results to illustrate the structure-activity relationship.
4. I noticed that there are only 59 references in this review, which is obviously not enough.
Reviewer 4 Report
Comments and Suggestions for Authors
Dear all,
Very intersting review about encapsulation, but some issues need to be dressed before the publication acceptance
1) add figures separtly to explain these parts (Diffusion; Erosion and degradation; Fragmentation) you refer to figue2, the reader should go back from figure 3 to see figure 2, is better to add others in the same pargraph
2) same remarques , you give an excellent figures to explain some encapsulation methodes and you stop drawing in (Covalent crosslinking; Polyelectrolyte crosslinking; Polysaccharide and drug conjugation; Self-assembly) please add figures and schemes to explain more these methodes!!!
3) same remarques : Hydrogel Additives; is better to add figures the reader will enjoy redeaning your review!!
4) Figures 16 ,18, 19, 26 and 27 are not so good!! please redraw
5) homogenize your references by following the same model
with regards
Reviewer 5 Report
Comments and Suggestions for Authors
The manuscript by Akpo E. et al, entitled: Encapsulation of Active Substances in Natural Polymer Coatings, is a very useful and interesting review of the encapsulation process of different active ingredients, highlighting the importance and properties of natural polymer coatings.
The review is well documented and systematized and also very clearly written.
Nevertheless, I would like to address some questions to the authors:
- Why did they choose to present only 3 of the encapsulation methods? I think that even if they are not described, all currently used methods should at least be mentioned in a tabular form. And would be better to split them into physico-chemical and mechanical technologies.
- It is surprising that the authors did not even mention chitosan as a natural polysaccharide, preferring instead to extend the presentation of arabic gum, which has the lowest utilization of them all. Chitosan is, currently, the most studied of all biopolymers used in the encapsulation process.
- Please, add a phrase explaining why you choose to limit only to natural polymers, taking into consideration that some very studied semisynthetic and synthetic polymers are also biocompatible and biodegradable.
Comments on the Quality of English Language
Minor Editing of English language is required
Round 2
Reviewer 1 Report
Comments and Suggestions for Authors
I must admit that the authors made substantial efforts to improve figures contents and references. The result is average.
Reviewer 2 Report
Comments and Suggestions for Authors
The authors have made the necessary modifications to their manuscript, making it suitable for publication.
Perhaps they could replace the current cyclodextrin structure with a truncated cone and place a Glcp unit on the side of the cone.
Reviewer 4 Report
Comments and Suggestions for Authors
Dear all,
Tha authors have performed all the requirements
with regards
Reviewer 5 Report
Comments and Suggestions for Authors
Thank you for the replies